# One Hundred Consecutive Neutropenic Febrile Episodes Demonstrate That CXCR3 Ligands Have Predictive Value in Discriminating the Severity of Infection in Children with Cancer

**DOI:** 10.3390/children10010039

**Published:** 2022-12-25

**Authors:** Małgorzata Nowak, Katarzyna Bobeff, Justyna Walenciak, Julia Kołodrubiec, Krystyna Wyka, Wojciech Młynarski, Joanna Trelińska

**Affiliations:** Department of Pediatrics, Oncology and Hematology, Medical University of Lodz, Sporna 36/50, 91-738 Lodz, Poland

**Keywords:** CXCR3 ligands, children, cancer, neutropenia, infection

## Abstract

This study assesses the value of the CXCR3 ligands CXCL9/MIG, CXCL10/IP-10 and CXCL11/I-TAC when used to supplement the standard infection markers C-reactive protein (CRP) and procalcitonin (PCT) in the diagnostic algorithm of neutropenic fever in children with cancer. The concentration of CRP, PCT and chemokines was determined during the first hour of fever and 12–24 h afterwards in pediatric oncology patients with neutropenia. Among 100 consecutive febrile episodes in neutropenic patients, 34 cases demonstrated fever of unknown origin (FUO) (group A), 47 demonstrated mild clinically or microbiologically proven infection (Group B) and 19 severe infection (Group C). Significantly higher PCT-1 levels were found in group C (0.24 ng/mL) vs. group A (0.16 ng/mL), and PCT-2 in group C (1.2 ng/mL) vs. A (0.17 ng/mL), and in C vs. B (0.2 ng/mL). Chemokine concentrations (I-TAC-1, IP-10-1, IP-10-2) were significantly lower in Group A vs. B+C; I-TAC 1: 48.64 vs. 70.99 pg/mL, *p* = 0.03; IP-10 1: 59.95 vs. 96.84 pg/mL, *p* = 0.04; and IP-10 2: 102.40 vs. 149.39 pg/mL, *p* = 0.05. The selected pro-inflammatory chemokines I-TAC and IP10 might help to distinguish cancer patients with febrile neutropenia with the highest risk of infection. Although procalcitonin could serve as a marker of a high risk of infection, its delayed response diminishes its usefulness.

## 1. Introduction

One of the major acute complications of intensive chemotherapy in children treated for cancer is infection, caused by inter alia damage to the mucosal barriers, insertion of a central vascular catheter and most commonly, neutropenia. Neutropenic infections are often asymptomatic, and the only symptom may be fever. Febrile neutropenia (FN) may occur due to non-infectious causes, such as drug administration, transfusions of blood products or neoplastic disease. However, it may be the first symptom of a severe life-threatening systemic infection such as sepsis. Currently, the standard procedure for treating patients with fever and neutropenia is based on hospitalization and immediate empirical treatment with intravenous broad-spectrum antibiotics, before confirming the presence of an infection [1]. This can be followed by antibiotic therapy in the case of infection; however, microbiological tests are first necessary to establish the etiology of the infection. It should be strictly performed before starting antibiotic therapy; however, a relatively long waiting time for the results means that they can only confirm the etiology of infection and indicate the optimal treatment based on the culture result [2,3,4].

Unfortunately, the biomarkers currently used in the diagnosis of neutropenic infection, such as C-reactive protein (CRP) and procalcitonin (PCT) are not satisfactory, and the diagnosis of FN hence presents a great challenge for clinicians [4]. Optimal inflammatory markers would allow low-risk patients to be differentiated from those at high risk of severe systemic infection. Although many potential biomarkers of infection have been studied, for example interleukin-6 (IL-6), interleukin-8 (IL-8), interferon gamma (TNF-α) and tumor necrosis factor alpha (TNF-β), satisfactory results have still not been obtained [4,5,6].

The search for suitable biomarkers has increasingly favored chemokines: small (~8–14 kDa), structurally cytokine-like, secreted proteins that regulate cell trafficking through interactions with a subset of 20 different seven-transmembrane, G protein-coupled receptors (GPCRs) [7]. The most-investigated chemokines belong to the CXC family. The CXCR3 receptor and its ligands represent a complex chemokine system whereby one receptor has three interferon (IFN)-ɣ inducible ligands: CXCL9 (also known as monokine induced by gamma interferon, MIG), CXCL10 (interferon-induced protein of 10 kDa, IP-10) and CXCL11 (interferon-inducible T-cell alpha chemoattractant, I-TAC) [8]. They are responsible for chemotaxis, cell migration and adhesion of immune cells to the site of infection [9].

Some promising data indicate an increase in chemokine level in children with sepsis, pneumonia and skin abscess, or septicemic and NEC cases in infants [10,11]. However, there are no relevant data on children with cancer.

We hypothesize that CXCR3 ligands used in combination with standard infection markers (CRP, PCT) would improve the diagnostic algorithm for children with neutropenic fever.

## 2. Materials and Methods

### 2.1. Study Population

This prospective study was conducted from May 2017 to August 2019 at the Department of Pediatrics, Oncology and Hematology, of the Medical University of Lodz, Poland. This study was approved by the Bioethics Committee of the Medical University of Lodz (#RNN/129/17/KE). It was conducted in compliance with good clinical practice guidelines and under the principles of the Declaration of Helsinki. All individuals, or their legal representatives, gave their written informed consent to take part.

This study included pediatric patients and young adults, i.e., aged 1–21 years, with neutropenia due to immunosuppressive anti-cancer therapy. The primary prophylaxis against infections included: trimethoprim–sulfamethoxazole (cotrimoxazole) for prophylaxis of Pneumocistis jirovecii pneumonia for all patients during immunosuppressive therapy, and oral Posaconazole or Amphotericin B for prophylaxis of invasive fungal infection for patients during induction therapy for acute leukemia.

When fever appeared, a physical examination was performed to assess the site of infection and blood samples were drawn for routine laboratory analysis with bacterial and fungal cultures. In addition, cultures from other sites (urine, stool, throat or wound smear) and radiological evaluation were performed when indicated. Subsequently, the patient received broad-spectrum intravenous antibiotics.

The concentrations of CRP and PCT, as standard infection markers, were assessed immediately (point 1: CRP 1, PCT 1) and re-evaluated 12–24 h from first collection (point 2: CRP 2, PCT 2). The remaining plasma materials from point 1 and point 2 were frozen and used to determine the concentration of chemokines CXCL9 (MIG), CXCL10 (IP-10) and CXCL11 (I-TAC).

The study group was divided into three groups according to the severity of infection: fever of unknown origin (FUO)—group A; clinically or microbiologically documented infections (with exception of bloodstream infection or sepsis)—group B; and bacteriemia, sepsis and patients transferred to Intensive Care Unit (ICU)—group C.

### 2.2. Definitions

Neutropenia was defined as an absolute neutrophil count (ANC) in peripheral blood less than 0.5 × 10^9^/l or an ANC ≤ 1.0 × 10^9^/l expected to decrease to an ANC less than 0.5 × 10^9^/l. Fever was defined as a single oral temperature of ≥38.3 °C, or temperature of ≥38.0 °C for ≥1 h [12]. Clinically documented infection (CDI) was diagnosed when a site of infection was documented but its microbiological pathogen either could not be proven or was inaccessible to examination. Microbiologically documented infection (MDI) was diagnosed when the infection was clinically detectable and microbiologically confirmed.

Bacteremia was recorded when a recognized pathogen was cultured from one or more blood cultures. Sepsis was diagnosed when symptoms of SIRS (systemic inflammatory response syndrome) occurred in the presence or as a result of suspected or proven infection. Finally, fever of unknown origin (FUO) was recorded in the case of fever without clinical or microbiological evidence of infection [13,14].

### 2.3. Laboratory Measurements

Concentration of C-reactive protein and procalcitonin were determined by standard laboratory methods in the hospital central laboratory. The other parameters were determined using ELISA: Interferon-inducible T-cell Alpha Chemoattractant—I-TAC (DCX110, R&D, Minneapolis, MN, USA), Interferon-gamma inducible protein 10 kD—IP10 (DIP100, R&D, Minneapolis, MN, USA) and Monokine Induced by Gamma Interferon—MIG (DCX900, R&D, Minneapolis, MN, USA).

### 2.4. Statistical Analysis

Statistical analyses were performed using Statistica software (Version 13; StatSoft, Inc., Tulsa, OK, USA). The distribution of the data was assessed with the Shapiro–Wilk test, and *p* > 0.1 was considered to indicate a normal distribution. In cases where the data were not normally distributed, i.e., where the data were skewed, nonparametric tests were used, with the median being used for comparisons instead of the mean.

Pairs of groups were compared using the Mann–Whitney U-test. Multiple groups were compared with the nonparametric analysis of variance (Kruskal–Wallis ANOVA); if ANOVA yielded a significant difference, this was followed by between-group multiple comparisons with a nonparametric post hoc test.

For all tests, *p* < 0.05 was deemed to be significant. However, for the Kruskal–Wallis nonparametric analysis, results with 0.05 < *p* < 0.1 were included.

Receiver operating characteristic (ROC) curve analyses were also performed at both time points. Threshold values were determined based on Youden’s index, and sensitivity and specificity were determined for each cut-off value.

The accuracy of the diagnostic tool was assessed based on the area under the curve (AUC) criterion. Following this, significant AUC values were compared with each other to distinguish the most applicable. The parameters demonstrating the best combination of sensitivity, specificity and accuracy were then used to distinguish the exposed (i.e., high risk of infection) from the high and low risk of severe infection. Patients considered to have only fever without any other symptoms of infection were placed in Group A, while those with more severe infection were allocated to Groups B and C. The negative and positive predictive values (NPV and PPV) were calculated, as well as the relative risk (RR) and odds ratio (OR).

## 3. Results

### 3.1. Patient Characteristics

One hundred consecutive febrile episodes in 62 patients, recorded between May 2017 and August 2019, were analyzed. The median age of the study group was 9.79 years (4.9–13.65). The basic characteristics of the infection episodes are presented in Table 1. The group included 34 febrile episodes in Group A, 47 episodes in Group B and 19 episodes in Group C. No significant difference in mean age, distribution of underlying disease duration of neutropenia or mean neutrophil count was observed between the groups.

Among Group B and Group C, the most frequently recorded infection was gastroenteritis and the most common detected pathogen was *Escherichia coli* ESBL (+). Among the bloodstream infections in Group C, Gram-positive bacteria were isolated in eight episodes and Gram-negative bacteria in five. Six patients were transferred to ICU due to septic shock (two patients), typhlitis (two patients), multiorgan failure (two patients). Mechanical ventilation was used in all ICU-patients. One patient died due to infection. Detailed data are presented in Appendix A.

### 3.2. Biomarker Analysis

The median values and interquartile ranges of the measured biomarkers at point 1 and point 2 are presented in Appendix A. No significant differences in CRP 1 or CRP 2 concentration were noted between groups (Figure 1a). Significantly higher concentrations of PCT 1 were found in Group C compared to Group A (median: 0.24 vs. 0.16 ng/mL, *p* = 0.05) and PCT 2 in Group C vs. A and C vs. B (median: 1.2 vs. 0.17 ng/mL, *p* = 0.006 and 1.2 vs. 0.2 ng/mL, *p* = 0.009) (Figure 1b). Lower values were noted for I-TAC 1 and IP-10 1 in Group A compared to Group B; however, this difference was not statistically significant. Significantly lower IP-10 2 concentrations were found in Group A compared to Group B (median: 102.40 pg/mL vs. 168.13 pg/mL, *p* = 0.006)—Figure 1c,d. MIG concentrations were comparable between groups A, B and C (Figure 1e).

After combining Group B with Group C, chemokine concentrations were found to be significantly lower in Group A vs. B+C: I-TAC 1: 48.64 vs. 70.99 pg/mL, *p* = 0.03; IP-10 1: 59.95 vs. 96.84 pg/mL, *p* = 0.04; and IP-10 2: 102.40 vs. 149.39 pg/mL, *p* = 0.05), Figure 2.

ROC curves were created for all analyzed parameters at both time points (Appendix A). The best cut-off values for distinguishing patients with a lower risk of infection (FUO, group A) from those with a higher risk (group B+C) were as follows: I-TAC 1 (<33.34 pg/mL) sensitivity 87.9% and specificity 41.2% (AUC 0.637, *p* = 0.02), IP-10 1 (<63.3 pg/mL) sensitivity 72.7% and specificity 55.9% (AUC 0.624, *p* = 0.003) and IP-10 2 (<129.9 pg/mL) sensitivity 57.6% and specificity 70.6% (AUC 0.623, *p* = 0.03) (Figure 3, Appendix A).

Assuming I-TAC 1 > 33.34 pg/mL as a cut-off value, 78 patients were indicated as being at high risk of severe infection, which actually occurred in 58 (74.4%) of cases, whereas only in 8 cases of the group with I-TAC 1 < 33.34 pg/mL. This resulted in OR of 5.08 (95% CI, 1.86–13.89) and RR of 2.04 (95% CI, 0.85–4.92) (Appendix A).

## 4. Discussion

Infections, especially bacterial sepsis, remain a major serious complication of neutropenia and constitute the most significant cause of mortality and morbidity in cancer patients undergoing intensive chemotherapy [15,16]. As such, there is a need for early markers that can determine which patients are at high risk for serious bacterial infection and need antibiotic therapy, and in whom the risk is low and antibiotics can be withheld.

Diagnosing bacterial infection in patients with febrile neutropenia is typically based on standard infection markers such as CRP and PCT. However, while CRP is highly sensitive, it is less specific for detecting bacterial infection [13]. In contrast, PCT has been found to be a more specific marker of bacterial infection and is slightly more pathogen dependent, especially in Gram-negative bacteremia [17,18].

Our present findings are consistent with previous literature data as the CRP concentration was not found to differ significantly between groups. Additionally, the highest PCT levels were noted in bloodstream infection and in patients transferred to ICU, indicating that PCT could be a marker of severe infection in cancer patients with neutropenia. However, it needs to highlighted that in patients with severe infection (Group C), median PCT level at fever onset was still within the normal range (below 0.5 ng/mL) and only rose to 1.2 ng/mL after 12–24 h. Similarly, previous studies report a PCT cut-off value of 0.25 ng/mL for distinguishing bacterial infection from other causes of fever [6,19].

As both CRP and PCT have some limitations, such as non-specificity and delayed response, it is imperative to discover new markers which could improve the diagnostic algorithm of neutropenic fever. Therefore, the present study evaluated the CXCR3 ligands CXCL9 (MIG), CXCL10 (IP-10) and CXCL11 (I-TAC) as candidates: the first such study in neutropenic patients. Among these candidates, CXCL9 (MIG) was not found to be a useful marker; however, IP-10 and I-TAC offered promise as possible diagnostic tools. A key finding was that both chemokines were present at lower levels in patients with FUO (Group A) compared to those with confirmed infection (Groups B+C) at fever onset. In addition, at the first time point, I-TAC with cut-off value 33.34 pg/mL demonstrated the highest sensitivity (87.9%); however, the specificity of 41.2% is quite low. IP-10 concentration taken at either time point also seemed to be effective at identifying patients with a low risk of infection.

We state that I-TAC is useful for extraction high risk infection patients due to its sensitivity in first time point. We are conscious that it is not a perfect diagnostic test because of its low specificity. However, it cannot be interpreted as a single diagnostic test. If ever, it always should be taken into consideration along with other inflammatory markers, and above all the patient’s condition.

Chemokines are mainly secreted at the site of inflammation by monocytes, and by epithelial and endothelial cells, and their levels can differ according to the severity and location of infection, as well as the nature of the pathogen [10]. Elevated plasma concentrations of MIG and IP-10 have been found in preterm infants with bacterial infection and in patients with visceral leishamiasis [11,20]. Additionally, IP-10 level was higher in patients with sepsis and in patients who required oxygen supplementation [10]. Although CXCR3 ligands have been studied in other conditions such as tuberculosis, HIV and interstitial pneumonia [21,22,23], the present study is the first to assess their diagnostic value in neutropenia. Our findings indicate that IP-10 and I-TAC may be effective markers for discriminating patients with microbiologically or clinically diagnosed infection from those without (FUO) in combination with standard laboratory parameters and patients’ individual risk factors.

This study has some limitations. No control group of healthy children was used, which could provide baseline levels for chemokines. In addition, the analysis did not include any viral or fungal infections; it would be valuable for future studies to include a wider range of fever origins, i.e., other than bacterial infections. Moreover, since in our statistical approach, we did not design any validation cohort, this disallows us to perform cross-validation of our results. This should be treated as a significant limitation. The cut-offs calculated using Youden index in ROC analysis refer to the study group only; hence, the findings need to be verified on an independent cohort of patients in the future.

## 5. Conclusions

The selected pro-inflammatory chemokines I-TAC and IP10 might help to distinguish cancer patients with febrile neutropenia with the highest risk of infection at fever onset. While procalcitonin could serve as marker of a high risk of infection, its delayed response diminishes its usefulness. 

## Figures and Tables

**Figure 1 children-10-00039-f001:**
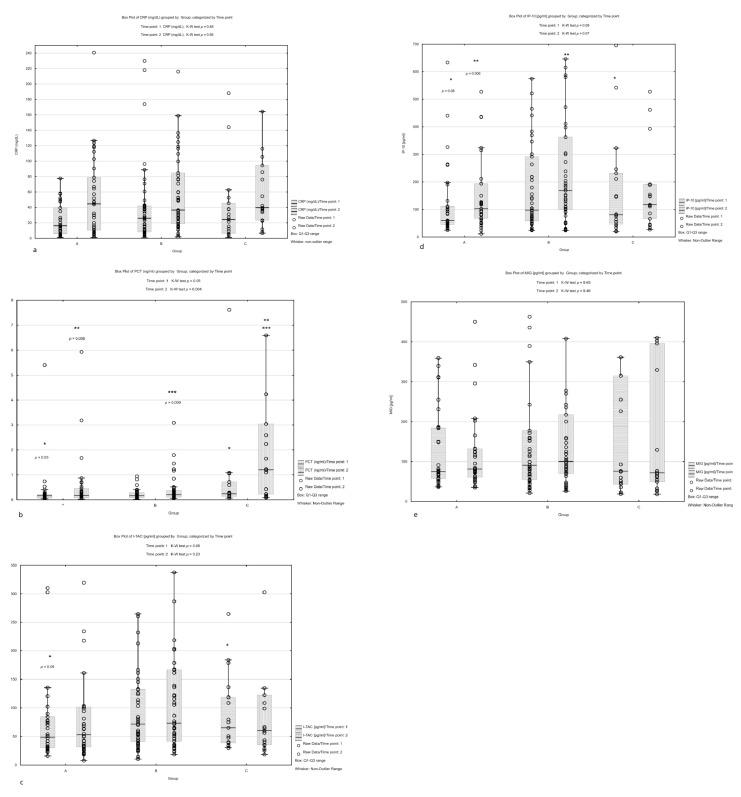
Comparisons of all markers’ concentrations between groups A, B and C.

**Figure 2 children-10-00039-f002:**
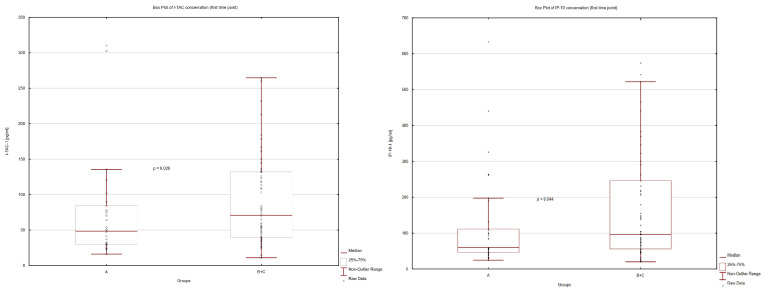
Comparisons of I-TAC-1 and IP-10-1 concentration between groups A and B+C.

**Figure 3 children-10-00039-f003:**
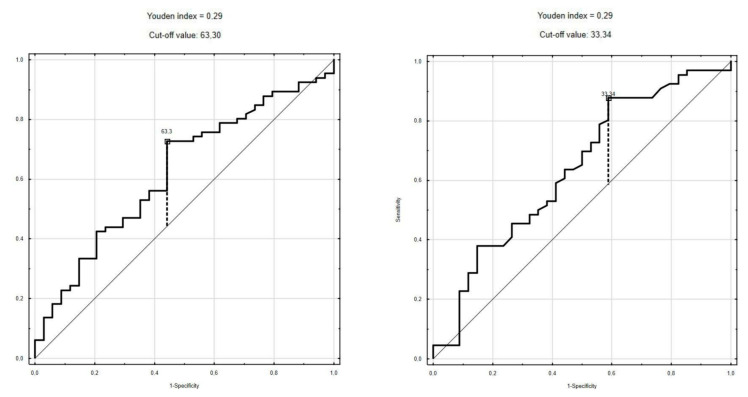
Receiver operating characteristics curves of I-TAC-1 and IP-10-1.

**Table 1 children-10-00039-t001:** Characteristic of the study group.

	All Episodes	Group A	Group B	Group C	*p* Value
	(n = 100)	(n = 34)	(n = 47)	(n = 19)	
Sex					0.49 *
n (% of female)	48 (48%)	19 (56%)	20 (43%)	9 (47%)	
n (% of male)	52 (52%)	15 (44%)	27 (57%)	10 (53%)	
Age (years)	9.8	9.8	10.5	7.53	0.48 **
Age (median [Q1–Q3])	(4.9–13.6)	(6.1–13.5)	(4.8–14)	(4.5–13.5)	
Underlying disease:					0.05 *
Leucaemias and Lymphomas	66	16	36	14	
- First line of treatment	58	15	31	12	
- After bone marrow transplantation	8	1	5	2	
Brain tumors	8	4	1	3	
Solid tumors	26	14	10	2	
Disease status:					0.66 *
Remission	85	30	40	15	
Relapse/Progression of malignancy	15	4	7	4	
Neutrophil count, mean ± SD (×10^3^/uL)	0.065	0.03	0.08	0.1	0.12 **
(median [Q1–Q3])	(0.02–0.17)	(0.02–0.1)	(0.03–0.23)	(0.02–0.13)	
Duration of neutropenia (days)	4 (2–6.5)	4 (2–6)	3 (4–13)	7 (2–6)	0.17 **
Fever at home	52	15	30	7	0.07 *
Inpatient fever	48	19	17	12	
Type of infection		FUO	MDI/CDI:	MDI/CDI:	
			Gastroenteritis—22	Bacteriemia—11	
			Mucositis—8	Sepsis—4	
			Pneumonia—7	Thyphilitis—2	
			Urinary tract infection—6	Multiorgan failure—2	
			Soft tissue		
			Infection—4		

* Pearson’s Chi-squared test. ** Kruskall–Wallis test. FUO—fever of unknown origin. MDI—microbiologically documented infection. CDI—clinically documented infection.

## Data Availability

All other data generated during the study are available from the corresponding author on reasonable request.

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
