# Peer review of "One Hundred Consecutive Neutropenic Febrile Episodes Demonstrate That CXCR3 Ligands Have Predictive Value in Discriminating the Severity of Infection in Children with Cancer"

_children, 2022, doi:10.3390/children10010039_

Round 1

Reviewer 1 Report

Nowak et al present data on 100 episodes of neutropenic fever from a single center study. The topic is interesting as there is a need to better discriminate severe FN from less severe episodes where antibiotics may be with held or stopped early. However, the manuscript could be further improved and some clarifications are needed.

Introduction:

OK

Material & methods:

1.     Clarify the routine management of FN at this hospital. It is stated that the first CRP was collected within the first hour of fever-where all children already admitted to a ward? Is there any primary profylaxis against bacterial, viral or fungal infections at your institute?

2.     Group C needs clarification; were the patients in ICU transferred directly to ICU when the fever appeared or did they belong in group A or B initially?

3.     Table 1; how did you classify the bacterias as gastroenteritis? Where these in fact stool samples only or did the patients also have symptoms with diarrhea? Or was this only colonizing flora?

4.     Please define sepsis vs septic shock. Which criteria did you use?

5.     Have you performed a power analysis?  If so, please report. Who provided the statistical expertise among authors? There are very many analysisis performed on a not so large cohort

6.     Motivate the use of Youden´s index. What should be considered a “good value” for a diagnostic test?

Results

1.     Table 1 is not helpful. I suggest to organize data so that we can compare group A,B and C for age, sex, underlying diseases and neutrophil counts as well as infection episodes. It would also be helpful if number of days with neutropenia in the reported episodes were provided. This is a useful marker for immunosuppression…

2.     Figure 1 should be improved. I suggest to include time-point 1&2 in the same panels for each marker. It will show the dynamics for each biomarkers between the groups.

3.     Please motivate combining group B+C  for further analyses

4.      

Discussion.

1.     The discussion lack in depth discussion on the current data compared with literature. Is the patient cohort representative?

2.     There needs to be a discussion on the low specificity for the suggested biomarkers. I disagree that I-TAC and IP-10 are good markers with such a low specificity…. I also wonder whether the study has enough power. Please provide more arguments

3.     Youden index of 0.29 -is that really a good diagnostic test?

Author Response

Dear Reviewer,

We greatly appreciate your interest in our manuscript and made our best effort to introduce the changes suggested during Your revision. The responses to each of Your concerns are listed below. We hope that the revised manuscript will prove to be of sufficient quality to consider its publication in your prestigious journal.

Joanna Trelinska M.D., Ph.D.

Manuscript ID: children-2000988 

Reviewer comments:

Nowak et al present data on 100 episodes of neutropenic fever from a single center study. The topic is interesting as there is a need to better discriminate severe FN from less severe episodes where antibiotics may be withheld or stopped early. However, the manuscript could be further improved and some clarifications are needed.

Introduction:

OK

Material & methods:

  1. Clarify the routine management of FN at this hospital. It is stated that the first CRP was collected within the first hour of fever-where all children already admitted to a ward? Is there any primary profylaxis against bacterial, viral or fungal infections at your institute?

Response:

In our institute the only primary prophylaxis against infections include: trimethoprim-sulfamethoxazole (cotrimoxazole) for prophylaxis of Pneumocistis jirovecii pneumonia for all patients during immunosuppressive therapy and oral Posaconazole or Amphotericin B for prophylaxis of invasive fungal infection for patients during induction therapy for acute leukemia. We have added that sentence to the text.

The routine management of FN in our institute was described in Method section: “When fever appeared, a physical examination was performed to assess the site of infection and blood samples were drawn for routine laboratory analysis with bacterial and fungal cultures. In addition, cultures from other sites (urine, stool, throat or wound smear) and radiological evaluation were performed when indicated. Subsequently, the patient received broad-spectrum intravenous antibiotics.”

  1. Group C needs clarification; were the patients in ICU transferred directly to ICU when the fever appeared or did they belong in group A or B initially?

Response:

The division into group A, B and C was made retrospectively, based on microbiological results, development of symptoms and necessity of ICU transfer. For clarification we have rewrite a sentence:

“The study group was divided into three groups according to the risk of severe severity of infection episode: fever of unknown origin (FUO) – group A; clinically or microbiologically documented infections (with exception of bloodstream infection or sepsis) – group B;  bacteriemia, sepsis and patients transferred to Intensive Care Unit (ICU) – group C.”

  1. Table 1; how did you classify the bacterias as gastroenteritis? Where these in fact stool samples only or did the patients also have symptoms with diarrhea? Or was this only colonizing flora?

Response:

Our definition for microbiologically-documented infection (MDI) was when the infection was clinically detectable and microbiologically confirmed. According to this definition we have diagnosed 16 episodes gastroenteritis microbiologically documented in patients with diarrhea and positive stool samples. Table S1 Supplementary files

According to definition of CDI (Clinically-documented infection was diagnosed when a site of infection was documented but its microbiological pathogen either could not be proven or was inaccessible to examination) we have diagnosed 6 gastroenteritis clinically documented in patients with diarrhea with negative stool samples. Table S1 Supplementary files

  1. Please define sepsis vs septic shock. Which criteria did you use?

Response:

To define sepsis and septic shock we have use the criteria according to Shankar-Hari et al. Jama 2016: Developing a new definition and assessing new clinical criteria for septic shock: for the third international consensus definition for sepsis and septic shock.

„Patients with septic shock can be identified using the clinical criteria of hypotension requiring use of vasopressors to maintain mean blood pressure of 65 mm Hg or greater and having a serum lactate level greater than 2 mmol/L persisting after adequate fluid resuscitation.”

  1. Have you performed a power analysis?  If so, please report. Who provided the statistical expertise among authors? There are very many analysisis performed on a not so large cohort

Response:

Statistical analysis was provided by Katarzyna Bobeff MD. We conducted post-hoc power analysis. The analysis revealed that power of our comparison is not enough due to high level of variance. Regarding that we decided to join groups B and C to enlarge number of patient with probable higher levels of chemokines and increase power of testing what resulted in detection of expected significant differences in I-TAC and IP-10 levels.

We are aware that there are many analyses, therefore non-parametric test and post-hoc for non-

parametric ANOVA were performed to avoid multiple testing and to minimize effect of lack of homogeneity of variance.

  1. Motivate the use of Youden´s index. What should be considered a “good value” for a diagnostic test?

Response:

We used Youden index method to describe the best cut-off value. Taking into consideration “good value” for diagnostic test, it should be >0,5 and at least AUC>0,7. Although we decided to present those results because of high sensitivity of certain chemokines, which we consider useful in clinical management of patients with neutropenia. 

Results

  1. Table 1 is not helpful. I suggest to organize data so that we can compare group A,B and C for age, sex, underlying diseases and neutrophil counts as well as infection episodes. It would also be helpful if number of days with neutropenia in the reported episodes were provided. This is a useful marker for immunosuppression…

Response:

We have change the Table 1 by adding some more clinical parameters: disease status, duration of neutropenia and place of fever onset (home/hospital).

  1. Figure 1 should be improved. I suggest to include time-point 1&2 in the same panels for each marker. It will show the dynamics for each biomarkers between the groups.

Response:

According to reviewer suggestion we have change Figure 1, including the both time-points in the same panel.

  1. Please motivate combining group B+C  for further analyses

Response:

Statistical analyses did not reveal expected effects due to lack of power of testing. We decided to increase power by enlarging group and resigning from multiple groups. Moreover, this operation has clinical implication of distinction the patient with high risk infection and those with low risk infection (FUO).  For clarification we have change the sentence:

“Patients considered to have only fever without any other symptoms be at low risk of infection were placed in Group A, while those with more severe infection were allocated to Group B and C.”

Discussion.

  1. The discussion lack in depth discussion on the current data compared with literature. Is the patient cohort representative?

Response:

Thera are so far no literature data concerning chemokines concentrations in children with cancer during infections episodes. That’s why we could not compare and discuss our data with the literature. However we have pointed that “elevated plasma concentrations of MIG and IP-10 have been found in preterm infants with bacterial infection and in patients with visceral leishamiasis. Additionally IP-10 level was higher in patients with sepsis and in patients who required oxygen supplementation”.

  1. There needs to be a discussion on the low specificity for the suggested biomarkers. I disagree that I-TAC and IP-10 are good markers with such a low specificity…. I also wonder whether the study has enough power. Please provide more arguments
  2. Youden index of 0.29 -is that really a good diagnostic test?

Response to point 2 and 3:

We state that I-TAC is useful for extraction high risk infection patients due to its sensitivity in first time point. We are conscious that it is not perfect diagnostic test because of its low specificity.
We absolutely agree that it cannot be interpreted as a single diagnostic test. If ever, it always should be taken into consideration along with other inflammatory markers and above all patient’s condition.
Nevertheless neutropenic fever protocols in children are implemented in every “suspicious” patient and its application probably means more careful vigilance of the high risk patient, not withdrawal of empiric therapy in those qualified as low risk.

Reviewer 2 Report

interesting findings, Thankyou for submitting the manuscript; Few minor comments; 

Introduction; In the first part of the introduction I would also add as reference the guideline of Lehrbecher and Sung  Guideline for the Management of Fever and Neutropenia in Children With Cancer and Hematopoietic Stem-Cell Transplantation Recipients: 2017 Update JCO 35, 18 2017 as this is an international accepted guideline on management of fever and neutropenia

 Methods; The study included pediatric patients and young adults, i.e. aged 1-21 years, with neutropenia due to immunosuppressive anti-cancer therapy

I would be interested in more parameters clinical according to the core clinical variable set; such as disease stastatus, chemointensity, prior stem cell transplant to decide eventually together with the lab parameters if dealing with high medium or low risk  (Some parameters are presented in table 1 but maybe you can extend this)

All lab parameters were performed on time point 1 (start of fever) and timepoint 2 12-24 hours later.

Was there a difference in time to presentation at the hospital Were all patients in patients? And if not is there a difference in time to presentation which might be of influence in your results, thinking of the importance of TTA (time to antibiotics)

 Discussion; IP-10 and I-TAC may be effective markers for discriminating pa-tients with microbiologically or clinically-diagnosed infection from those without (FUO). Yes I agree but I would always mention here together with other lab parameters and in combination with the clinical set of variables see Haeusler et al Curr Opin Infect Dis 2015, 28:532–538

 Conclusions; I would be a bit more cautious in your conclusions, as there are some limitations to the study 

Author Response

Dear Reviewer,

We greatly appreciate your interest in our manuscript and made our best effort to introduce the changes suggested during Your revision. The responses to each of Your concerns are listed below. We hope that the revised manuscript will prove to be of sufficient quality to consider its publication in your prestigious journal.

Joanna Trelinska M.D., Ph.D.

Manuscript ID: children-2000988 

Reviewer comments:

Comments and Suggestions for Authors

interesting findings, Thank you for submitting the manuscript; Few minor comments; 

Introduction; In the first part of the introduction I would also add as reference the guideline of Lehrbecher and Sung  Guideline for the Management of Fever and Neutropenia in Children With Cancer and Hematopoietic Stem-Cell Transplantation Recipients: 2017 Update JCO 35, 18 2017 as this is an international accepted guideline on management of fever and neutropenia

Response:

Thank you for pointing out the importance of international guidelines for the management of fever in children with cancer published by Lehrbecher et al. JCO 2017. According to the suggestion we have add this reference in to the introduction.

Methods; The study included pediatric patients and young adults, i.e. aged 1-21 years, with neutropenia due to immunosuppressive anti-cancer therapy

I would be interested in more parameters clinical according to the core clinical variable set; such as disease stastatus, chemointensity, prior stem cell transplant to decide eventually together with the lab parameters if dealing with high medium or low risk  (Some parameters are presented in table 1 but maybe you can extend this)

Response:

We have change the Table 1 by adding some more clinical parameters: disease status, duration of neutropenia and place of fever onset (home/hospital).

All patients were during intensive period of chemotherapy when fever occurred.

All lab parameters were performed on time point 1 (start of fever) and timepoint 2 12-24 hours later.

Was there a difference in time to presentation at the hospital Were all patients in patients? And if not is there a difference in time to presentation which might be of influence in your results, thinking of the importance of TTA (time to antibiotics)

Response:

We have added data where the fever started to the Table 1. We did not observe any statistically significant differences between groups: fever started at home vs at the hospital.

 Discussion; IP-10 and I-TAC may be effective markers for discriminating patients with microbiologically or clinically-diagnosed infection from those without (FUO). Yes I agree but I would always mention here together with other lab parameters and in combination with the clinical set of variables see Haeusler et al Curr Opin Infect Dis 2015, 28:532–538

Response:

According to reviewer suggestion we have change the sentence in the discussion:

“Our findings indicate that IP-10 and I-TAC may be effective markers for discriminating patients with microbiologically or clinically-diagnosed infection from those without (FUO) in combination with standard laboratory parameters and patients individual risk factors.”

Conclusions; I would be a bit more cautious in your conclusions, as there are some limitations to the study 

Response:

According to reviewer suggestion we have change the sentence in the conclusion:

“The selected pro-inflammatory chemokines I-TAC and IP10 might help to distinguish cancer patients with febrile neutropenia with the highest risk of infection at fever onset.”

Round 2

Reviewer 1 Report

The papers has improved after revision. The discussion is more in line with the data presented.

Author Response

Response to Reviewer comment:

We are grateful for the effort made by the Reviewer in revision of our manuscript. We did the best to improve it according to Reviewer suggestions.
